# Antimicrobial Effects of Some Natural Products on Adhesion and Biofilm Inhibition of *Clostridioides difficile*

**DOI:** 10.3390/pharmaceutics16040478

**Published:** 2024-03-30

**Authors:** Dorota Wultańska, Michał Piotrowski, Hanna Pituch

**Affiliations:** Department of Medical Microbiology, Medical University of Warsaw, 02-004 Warsaw, Poland; piotrowski.michal90@gmail.com (M.P.); hanna.pituch@wum.edu.pl (H.P.)

**Keywords:** *Clostridioides difficile*, *Clostridium difficile*, antibiofilm, antibacterial activity, plant bioactive products

## Abstract

Understanding the potential antimicrobial properties of natural compounds and their impacts on *Clostridioides difficile* virulence factors may aid in developing alternative strategies for preventing and treating *C. difficile* infections (CDI). In this study, we investigated the bactericidal effects of ginger oil (GO), peppermint oil (PO), curcumin (CU), cinnamon aldehyde (CI), and trans-cinnamaldehyde (TCI) on the adhesion and biofilm disruption of *C. difficile*. We used three reference and five clinical *C. difficile* strains of different ribotypes. The bactericidal activity was assessed using the broth microdilution method. The adhesion was evaluated using human epithelial cell lines, and biofilm formation was visualized by confocal laser scanning microscopy. All tested strains exhibited susceptibility to CU, with minimum inhibitory concentration (MIC) values ranging from 128 µg/mL to 2048 µg/mL. Similarly, all strains were susceptible to CI and TCI, with MIC values ranging from 6.25% (*v*/*v*) to 25% (*v*/*v*). Most of the tested substances reduced the adhesion of *C. difficile* strains, while two tested strains showed significantly higher adhesion when co-incubated with the tested substances. Similar observations were made for biofilm formation, with observed density and morphology varied depending on the strain. In conclusion, the tested products demonstrated bactericidal activity and reduced the adhesion of *C. difficile* strains. They may be considered for further studies as potential antimicrobial agents targeting biofilm-related infections.

## 1. Introduction

*Clostridioides difficile*, formerly known as *Clostridium difficile*, is a Gram-positive, spore-forming bacterium recognized as the primary causative agent of antibiotic-associated diarrhoea and pseudomembranous colitis [1]. This pathogen poses a significant healthcare burden, with increasing rates of infections, recurrences, and the emergence of hypervirulent strains. The dominant hypervirulent strains belong to polymerase chain reaction ribotype (RT) 027, which spreads across North America and numerous European countries [2,3]. Cases of *C. difficile* infections (CDIs) caused by RT027 were also documented in Poland [4,5]. Research in various countries has revealed that RT027 is linked to a global surge in hospital-related outbreaks, marked by recurring infections and a notably high mortality rate [6,7].

In addition to toxin production, biofilm formation by *C. difficile* is a crucial factor in bacterial pathogenicity and persistence [8,9,10]. Biofilms are structured communities of bacteria embedded within a matrix of extracellular polymeric substances, which may include proteins, polysaccharides, and DNA [8,9,10]. Biofilms promote the adhesion of *C. difficile* to the gastrointestinal lining. The extracellular polymeric substance matrix provides a stable surface for the bacteria to attach to, enhancing their ability to colonize and persist in the host’s gut [11]. The increased adhesion contributes to the overall virulence of *C. difficile*. Biofilms act as protective shields for the bacteria, making them highly resistant to antibiotic treatments. Moreover, they play a key role in recurrent CDI [11]. Understanding the significance of *C. difficile* biofilms is crucial for developing more effective strategies to prevent and treat CDIs. Researchers continue to explore the methods of disrupting or inhibiting biofilm formation as well as to develop novel therapies that can target bacteria within biofilms or prevent bacterial adhesion [10].

The current guidelines recommend the use of fidaxomicin and vancomycin for the treatment of CDI [12]. However, traditional antibiotic therapies have not succeeded in completely eradicating CDI, often leading to recurrent disease and the emergence of antibiotic-resistant strains [13]. Consequently, there is a growing interest in exploring alternative approaches to combat CDI [14].

Natural products, such as curcumin (CU), peppermint oil (PO), ginger oil (GO), cinnamon aldehyde (CI), and trans-cinnamaldehyde (TCI), have demonstrated antimicrobial properties against various bacterial pathogens [15]. Curcumin, extracted from the rhizome of *Curcuma longa*, has been extensively studied for its anti-inflammatory and antibacterial effects [15,16]. The chemical composition of PO varies depending on the extraction method, environmental conditions, and geographical origin. Generally, its major constituents include menthol, epoxyocimene, linalool, menthone, eucalyptol, and neo-menthol [17]. The compounds present in cinnamon, CI, and TCI were reported to possess antibacterial activity. Additionally, GO contains bioactive compounds known for their potential health benefits [18,19,20]. The most abundant ingredients in ginger are α-Zingiberene (22.2%), β-Sesquiphellandrene (11.1%), 1,8-Cineole (6%), Geranial (5.1%), and β-Bisabolene (4.9%) [21].

In this study, we investigated the bactericidal effects of GO, PO, CU, CI, and TCI against *C. difficile*. We also examined the impact of these natural products on the adhesion of *C. difficile* to human colonic epithelial cells in vitro and their ability to disrupt *C. difficile* biofilm. Importantly, in addition to reference strains, we also attempted to test clinical hypervirulent strains as well. Understanding the potential antimicrobial properties of these natural products and their impact on *C. difficile* virulence factors could provide valuable insights and help develop alternative strategies for the prevention and treatment of CDIs.

## 2. Materials and Methods

### 2.1. Bacterial Strains

For this study, we used eight bacterial strains, which included reference strains 630 (RT012; TcdA^+^ TcdB^+^ CDT^−^), ATCC 9689 (RT001; TcdA^+^ TcdB^+^ CDT^−^), and M120 (RT078; TcdA^+^ TcdB^+^ CDT^+^). Additionally, we used five randomly selected clinical strains of *C. difficile*: 4308/13 (RT027), 25694/12 (RT023), 2628/12 (RT176), CD 15 (RT046), and 1128/06 (RT017) belonging to different RTs. These clinical *C. difficile* strains were isolated from stool samples from patients with antibiotic associated diarrhoea. Isolation of *C. difficile* was performed on selective Columbia Agar supplemented with 100 mg of cycloserine/L, 8 mg of cefoxitine/L, and 2 mg of amphotericin B/L (CCCA medium; bioMerieux, Marcy-l’Etoile, France). Plates were incubated in an anaerobic condition using Genbag and Genbox anaer gas generators (bioMerieux, Marcy l’Etoile, France) at 37 °C for 48 h. The isolates were identified as *C. difficile* by characteristic colony morphology, specific horse odour, yellow-green fluorescence under UV light (365 nm), and Gram staining, and isolates were confirmed as *C. difficile* via mass spectrometry (Vitek MS bioMerieux, Marcy l’Etoile, France). Tested strains were kept at −70 °C using the Microbank™ system (Pro-Lab Diagnostics, Bromborough, Wirral, UK). Before their use, we thawed strains and cultured them on Columbia agar plates with 5% sheep blood (Beckton Dickinson, Franklin Lakes, NJ, USA). Strains were incubated in anaerobic conditions at 37 °C for 48 h. Subsequently, all isolates were cultured in brain–heart infusion medium (BHI; Difco, Detroit, MI, USA) at 37 °C for 48 h under anaerobic conditions.

Strains 630 and M120 were generously provided by Professor Brendan Wren from the Department of Pathogen Molecular Biology at the London School of Hygiene and Tropical Medicine, London, UK. Strain ATCC 9689 was acquired from bioMérieux (Marcy l’Etoile, France). 

### 2.2. Tested Substances

The following substances of natural origin were used for the study: GO (ginger oil natural, FCC, FG, Sigma Aldrich, St. Louis, MO, USA), PO (peppermint oil natural, *Mentha piperita* L., Sigma Aldrich, USA), CU (*Curcuma longa* powder, Sigma Aldrich, USA), CI (Cinnamaldehyde natural, ≥95%, FG, Sigma Aldrich, USA), and TCI (trans-Cinnamaldehyde, ≥99%, Sigma Aldrich, USA). The stock solutions of liquid substances (50% *v*/*v*) were prepared using BHI medium, whereas CU was dissolved in BHI medium at an initial concentration of 4096 µg/mL.

### 2.3. Antimicrobial Effect of Tested Substances on C. difficile Planktonic Growth

The minimum inhibitory concentration (MIC) and minimum bactericidal concentration (MBC) values were determined for all tested substances. The MIC values were determined using the microdilution method, with the substances being diluted in BHI medium in titration plates. The following concentrations of GO, PO, CI, and TCI were tested: 50%, 25%, 12.5%, 6.25%, 3.12%, and 1.56% (*v*/*v*). Curcumin was tested at concentrations of 2, 4, 8, 16, 32, 64, 128, 256, 512, 1024, 2048, and 4096 µg/mL.

Wells containing 180 μL of each dilution were inoculated with a 20 μL suspension of *C. difficile* strains standardized to a 3 McFarland turbidity and incubated at 37 °C for 48 h under anaerobic conditions. The positive control consisted of BHI medium with 20 μL of a suspension of *C. difficile* strains with a turbidity of 3, while the negative control was BHI medium.

The MBC values were determined by plating 20 µL onto Columbia agar with 5% sheep blood (Becton Dickinson, GmbH, Franklin Lakes, NJ, USA) at individual concentrations, followed by incubation for 48 h under anaerobic conditions. Following incubation, *C. difficile* growth was assessed, and the first concentration at which there was no bacterial growth was recorded as the MBC value. The assay was performed in triplicate for each dilution.

### 2.4. Cell Cultures

In this study, three different human epithelial cell lines were employed, including HT-29 cells and mucus-secreting HT-29 MTX cells obtained from the European Collection of Authenticated Cell Cultures (ECACC, Salisbury, UK), as well as Human colon CCD 841 CoN cells sourced from the American Type Culture Collection. The maintenance conditions for these cell lines were consistent with those previously described [22,23]. The experiments were conducted on mature cells, specifically 15 days after seeding for HT-29 and CCD 841 CoN cells and 21 days after seeding for HT-29 MTX cells.

### 2.5. Effect of Tested Substances on C. difficile Adhesion

The adhesion assay used in this study was described in detail in our previous research [22,23]. In short, mature cells were grown in 24-well plates and washed twice with phosphate-buffered saline. Subsequently, cells were incubated for 4 h with Dulbecco’s Modified Eagle Medium (DMEM; Lonza, Walkersville, MD, USA) lacking antibiotic/antimycotic solutions, supplemented with a final concentration of 1% of either CU, CI, TCI, and GO, and PO was used at a concentration of 0.5%. A concentration of 1% of the substance was chosen for the adhesion test based on literature reports and our previous studies. For peppermint oil, a concentration of 0.5% was used because at a concentration of 1%, the substance had a toxic effect on eukaryotic cells. Following incubation, 100 μL of a *C. difficile* inoculum adjusted to a McFarland standard of 3 was introduced into each well and left to incubate for 1 h. Subsequently, the medium was aspirated, and the wells were rinsed twice with phosphate-buffered saline. The cells were then trypsinised for 10 min at 37 °C, and 500 μL of fresh media containing 10% fetal bovine serum was added to neutralize the trypsin. The contents of each well were transferred to sterile Eppendorf tubes, diluted tenfold, and 20 μL of the diluted solution was used to inoculate Columbia agar supplemented with 5% sheep blood. These plates were incubated under anaerobic conditions at 37 °C for 48 h. Every dilution was seeded in duplicate, and each assay was conducted in triplicate. Subsequently, colonies were enumerated, and the mean count was determined. Then, the adhesion coefficient was calculated for each measurement using the following formula: $$Adhesion\left(\%\right)=\frac{bacterial\;count\;in\;sample}{bacterial\;count\;in\;control}\times 100$$

### 2.6. Effect of Tested Substances on C. difficile Biofilm Formation on Confocal Laser Scanning Microscopy

Confocal laser scanning microscopy was performed according to the protocol described in previous studies [22,23]. Overnight cultures of all tested *C. difficile* strains were grown on sterile 10 mm diameter glass-bottom dishes (Nunc, Roskilde, Denmark) in BHI medium supplemented with 0.1 M glucose, with or without the presence of a sub-inhibitory concentration of the tested substances, which was half of the obtained MIC.

Biofilms were allowed to develop for 48 h at 37 °C under anaerobic conditions.

After reaching maturity, the biofilms were subjected to two washes using 10 mM MgSO_4_. Subsequently, acridine orange staining (10 μg/mL) was applied for 30 min in the dark. Following staining, the dishes were washed twice with 10 mM MgSO_4_. Imaging was performed using a Nikon A1R MP microscope with a Nikon Ti Eclipse series (Nikon, Tokyo, Japan) equipped with a ×60 objective lens and immersion oil. Images were acquired at a resolution of 2040 × 2048 pixels with a Z-step of 0.1 μm. Acridine orange fluorescence was detected using an excitation wavelength of 488 nm and an emission wavelength of 500 to 550 nm. Subsequently, image processing and analysis were performed using the NIS-Elements AR v. 4.10 software.

### 2.7. Statistical Analysis

The MIC and MBC values were analysed descriptively. The effect of tested substances on *C. difficile* adhesion was assessed using the two-way analysis of variance followed by Tukey’s post hoc test. Values of *p* less than 0.05 were considered statistically significant. All calculations were performed using the Statistica software (version 13, StatSoft, Kraków, Poland) and Prism (version 9.4.1, GraphPad Software, La Jolla, CA, USA).

## 3. Results

The antimicrobial effect of GO, PO, CU, CI, and TCI is presented in Table 1. All tested strains were susceptible to CU, with the MIC ranging from 64 µg/mL to 2048 µg/mL and the MBC ranging from 128 µg/mL to 4096 µg/mL. Additionally, all strains were susceptible to CI and TCI, with MIC and MBC values ranging from 6.25% (*v*/*v*) to 25% (*v*/*v*). In terms of susceptibility to PO, two strains (630 and M120) showed the MIC and MBC higher than 50% *v*/*v*, while in the other susceptible strains, MIC and MBC values ranged from 12.5% (*v*/*v*) to 50% (*v*/*v*). Finally, the MIC and MBC of GO were higher than 50% *v*/*v* in strains 630, M120, and 25694/12, while in the other susceptible strains, MIC and MBC values ranged from 6.25% (*v*/*v*) to 50% (*v*/*v*).

### 3.1. Effect of Tested Substances on C. difficile Adhesion

The effect of GO, PO, CU, CI, and TCI on *C. difficile* adhesion is presented in Figure 1. Detailed pairwise comparisons are included in the Appendix A.

For *C. difficile* strain 630, GO, CU, and TCI were found to increase its adhesion to CCD841 CoN cells, with TCI also increasing adhesion to HT-29 cells. However, all substances were highly effective in inhibiting this strain’s adhesion to HT-29 MTX cells. A similar pattern of inhibition was observed for strains ATCC 9689 and 4308/13, where all substances significantly reduced adhesion to all tested cell lines, with the exception of CI’s effect on HT-29 cells.

The study revealed varying levels of effectiveness across different strains and cell lines. For example, strain M120 showed the least reduction in adhesion; CU was the only substance to significantly reduce its adhesion across all cell lines, with GO and PO specifically targeting HT-29 MTX cells.

Strain 25694/12’s adhesion to CCD 841 CoN cells was notably reduced by CU, while all substances, except for TCI, significantly lowered its adhesion to both HT-29 and HT-29 MTX cell lines. For strain 2628/12, all tested substances, apart from TCI, effectively reduced adhesion to CCD 841 CoN and HT-29 MTX cells, with only CU and TCI impacting adhesion to HT-29 cells.

Furthermore, the substances generally inhibited the adhesion of strain CD15 to all cell types, except TCI’s effect on HT-29 MTX cells. Lastly, the adhesion of strain 1128/06 was significantly reduced by all substances, except for GO’s effect on CCD 841 CoN cells and TCI’s effect on HT-29 cells. This comprehensive analysis highlights the potential of these substances in preventing the adhesion of *C. difficile* to host cells, which is crucial for managing infections.

### 3.2. Effect of Tested Substances on Biofilm Formation by C. difficile

The results of the confocal laser scanning microscopy are presented in Figure 2 and Figure 3. The interpretation is included in Table 2. 

Following the use of CI and TCI, the confocal images of the biofilm became challenging to interpret due to artifacts resulting from the chemical reactions of these products with the dishes used to cultivate the biofilm. However, this issue was observed only for CI and TCI (Figure 2E,F,K,L,Q,R,W,X). 

## 4. Discussion

The increasing challenge of antibiotic resistance has revived interest in the antimicrobial properties of natural substances that have been used for centuries. Before the discovery of modern antibiotics, various cultures relied on the therapeutic properties of plant-derived natural products. Today, as antibiotic resistance complicates the treatment of infectious diseases, investigators and healthcare professionals continue to explore the potential use of natural substances. Due to their rich pharmacological diversity and minimal potential for resistance, they hold promise for developing novel treatments. Products such as honey, garlic, or plant extracts have shown substantial antimicrobial potential, offering hope for an alternative to conventional antibiotics [23,24,25,26]. By harnessing the power of these natural substances and incorporating them into modern medicine, we may find sustainable solutions to counter the growing antibiotic resistance crisis, preserving the efficacy of antimicrobial treatments for generations to come. In our study, we assessed the antimicrobial activity of GO, PO, CU, CI, and TCI against various RTs of *C. difficile*. Additionally, we investigated the impact of these substances on the adhesion and biofilm formation of *C. difficile.*

Ginger oil has a strong inhibitory effect on *Staphylococcus aureus*, *S. epidermidis*, *Escherichia coli* and *Pseudomonas aeruginosa.* Gram-positive bacteria were more affected than Gram-negative bacteria [20,27]. Natural essential oils with well-established antimicrobial potential were also shown to be effective against *Clostridium* spp. inhibition (*C. butyricum*, *C. intestinalis*, *C. hystoliticum*, *C. perfringens*, and *C. ramosum*), but they have not yet been tested against *C. difficile* [28]. GO directly targets the cell membrane, disrupting its structure and increasing its permeability. This leads to the loss of the bacteria’s essential structural functions, culminating in bacterial cell death at certain concentrations. Additionally, the hydrophobic compounds in GEO may interact with the lipophilic parts of the membrane and isolated mitochondria, compromising their integrity and functionality [20]. In our study, GO inhibited *C. difficile* strains of RT176, RT046, and RT017, with an MIC value of 12.5%, 12.5%, and 6.25%, respectively. Ginger oil significantly inhibited adhesion to cell lines, except the *C. difficile* strain RT017, which did not show a significant effect for the CCD841 CoN line. In the presence of GO, *C. difficile* formed a thin but regular biofilm, while planktonic cells underwent a morphological change and became rounded.

Peppermint essential oil exhibits potent bactericidal effects on *E. coli*, disrupting the integrity of bacterial cell membranes [29]. It also demonstrates similar effects against *S. aureus*, *Streptococcus faecalis*, *Bacillus subtilis*, *Neisseria gonorrhoeae*, and *P. aeruginosa* [30]. The antibacterial effect of PO is attributed to its antioxidant and enzyme inhibitory activities [29]. In our study, PO showed the most potent antimicrobial and anti-adhesive activity against the strain belonging to RT027. On the other hand, it had a mild impact on the biofilm. This may be due to the fact that it was used at a lower concentration of 0.5%, because higher concentrations are potentially detrimental to colon cells. Interestingly, the MIC and MBC of PO against reference strains 630 and M120 exceeded 50%. Moreover, PO did not decrease the adhesion of these strains to CCD 841 CoN and HT-29 cells; on the contrary, their adhesion appeared to be increased.

Curcumin has an inhibitory effect on various Gram-negative and Gram-positive bacteria. It also shows antibacterial activity against multidrug-resistant isolates [31]. In a recent study by Neto et al., the MIC values of CU against clinical isolates of methicillin-resistant *S. aureus* ranged from 125 to 500 μg/mL [32]. Sharahi et al. reported that the MIC values of CU against multidrug-resistant *A. baumannii*, *P. aeruginosa*, and *Klebsiella pneumoniae* range from 128 to 512 µg/mL [33]. Studies showed that CU can damage the permeability and integrity of the bacterial membrane, leading to bacterial cell death. Curcumin also serves as an immunomodulator, enhancing the body’s response to bacterial infections by inhibiting the pathogen’s virulence factors and boosting the host’s immune defence [31]. In several studies, CU was reported to inhibit bacterial quorum sensing systems/biofilm formation and prevent bacterial adhesion to host receptors in various species, including *S. aureus*, *E. faecalis*, *E. coli*, *Streptococcus mutans*, *Listeria monocytogenes*, *H. pylori*, *P. aeruginosa*, *Serratia marcescens*, *Aeromonas hydrophila*, and *A. baumannii* [31]. CU was shown to promote lactate dehydrogenase (LDH) production in *P. aeruginosa*, *S. aureus*, *and E. faecalis*, with the CU/LDH complex exhibiting antibacterial and antibiofilm effects [34]. Our study showed significant antibacterial activity of CU against *C. difficile*, with MIC values ranging from 64 mg/L to 2048 mg/L. Curcumin significantly inhibited the adhesion of *C. difficile* strains to all cell lines. The strains produced thick biofilm with high 3D architecture. Importantly, *C. difficile* planktonic cells changed their shape under the influence of CU.

Both CI and TCI exhibited the most potent antimicrobial properties against the tested strains, with MIC values ranging from 6.25% to 12.5% *v*/*v*. Of note, both substances displayed the lowest MIC of 6.25% *v*/*v* against the strains belonging to RT027. Both substances had similar anti-adhesive properties, except that TCI increased the adhesion of reference strain 630 to CCD 841 CoN and HT-29 cells. The images of the biofilm subjected to these substances were difficult to interpret. Roshan et al. confirmed the antimicrobial activity of cinnamon powder and TCI, with MICs of 75 mg/mL and 0.2 mg/L, respectively [35]. The antimicrobial action of cinnamon derivatives is based on several mechanisms, including alterations in cell membranes and the lipid profile of Gram-negative bacteria [36,37], as well as the inhibition of bacterial ATPase [37,38], cell division [37,39], and membrane porins [37,40]. Importantly, as shown in our study, CI and TCI decreased *C. diffcile* adhesion, which is the first step of adhesion. Amalaradjou et al. [40] assessed the impact of sub-inhibitory levels of TCI on *Cronobacter sakazakii*. They revealed a significant reduction both in motility and biofilm formation. The findings from reverse transcriptase quantitative polymerase chain reaction analysis confirmed that TCI downregulated flagellar motility genes (including *fliD*, *flgJ*, *motA*, and *motB*), leading to a reduction in biofilm formation.

The strength of our study is the fact that we used clinical isolates obtained from patients with CDI in addition to reference strains. 

Our study does have limitations, one of which includes the absence of biofilm assays on titration plates. Initially, we attempted to observe biofilm formation by cultivating it on titration plates. However, this approach proved to be unproductive as the substances tested interacted adversely with the material of the titration plates, exacerbating the effects. This interaction may have compromised the accuracy of our observations regarding biofilm formation, thus limiting the comprehensiveness of our results in this specific area.

## 5. Conclusions

In summary, our study has investigated the antimicrobial effects of natural products on various strains of *C. difficile*. The tested products, including GO, PO, CU, CI, and TCI, displayed significant bactericidal activity, particularly against hypervirulent strains, and effectively reduced the adhesion of *C. difficile*. The direction for future research should focus on further elucidating the mechanisms of action of these natural substances and exploring their integration into existing treatment paradigms, paving the way for more comprehensive strategies in the fight against CDI.

## Figures and Tables

**Figure 1 pharmaceutics-16-00478-f001:**
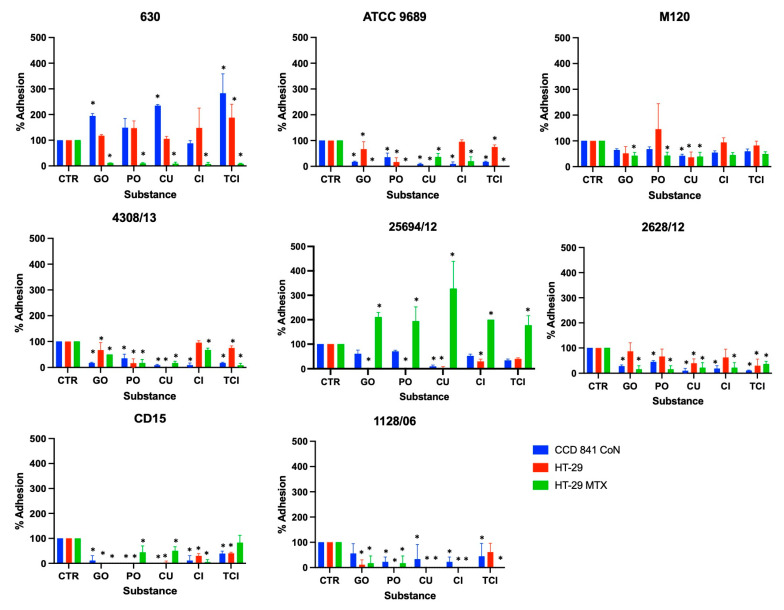
Effect of tested substances on *C. difficile* adhesion to three different cell lines. Error bars represent standard deviations; * indicates statistical significance *p* < 0.05. CTR—positive control; GO—ginger oil; PO—peppermint oil; CU—curcumin; CI—cinnamaldehyde; TCI—trans-cinnamaldehyde.

**Figure 2 pharmaceutics-16-00478-f002:**
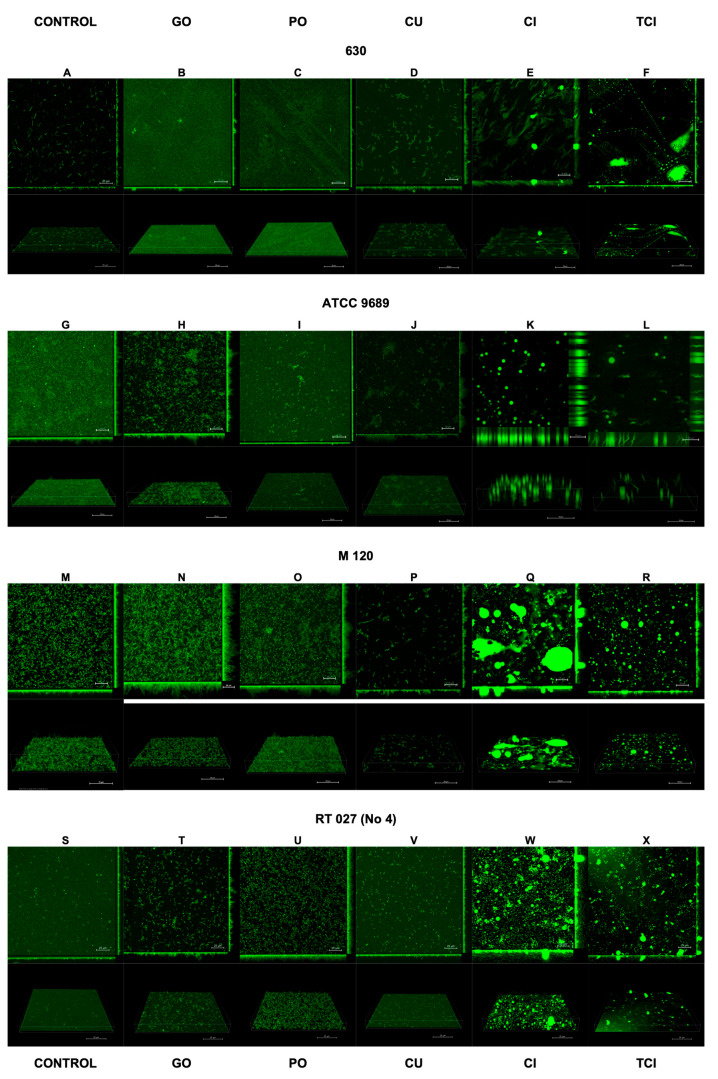
Effect of tested substances on *C. difficile* (strains 1–4) biofilm formation. Representative confocal microscopy images of horizontal (xy) and vertical (xz) projections of *C. difficile* biofilm structures. Slices viewed with maximum intensity projection. (**A**–**X**) are consequent images.

**Figure 3 pharmaceutics-16-00478-f003:**
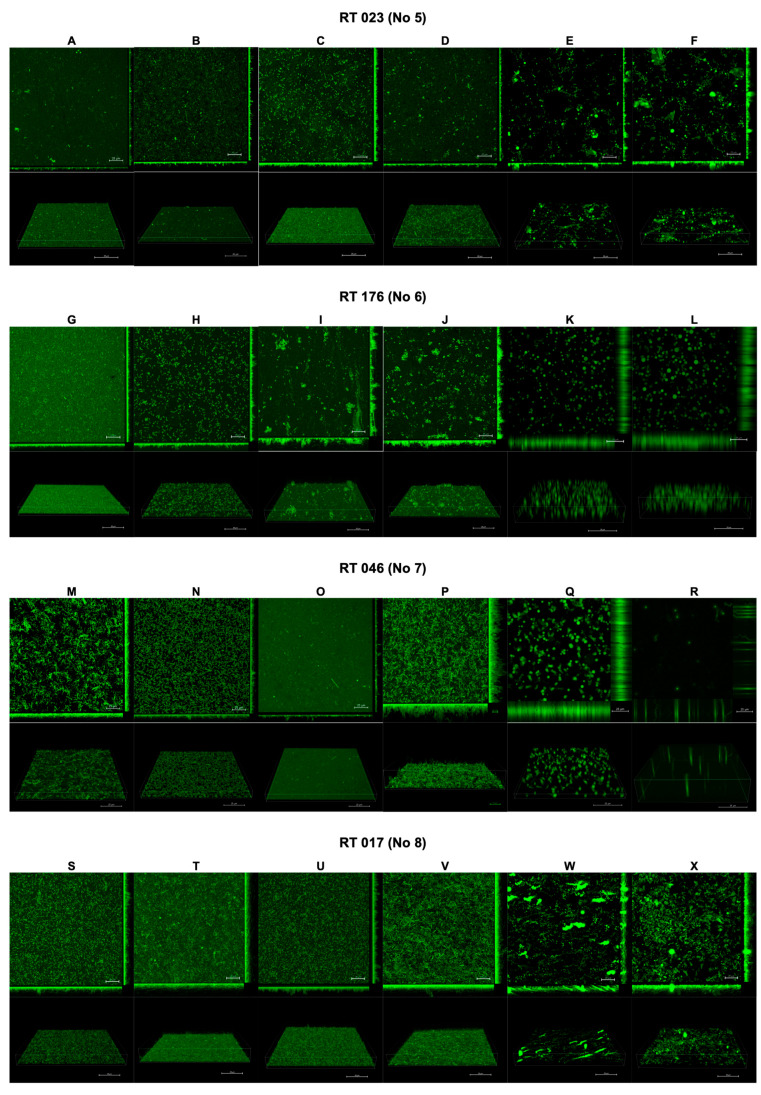
Effect of tested substances on *C. difficile* (strains 5–8) biofilm formation. Representative confocal microscopy images of horizontal (xy) and vertical (xz) projections of *C. difficile* biofilm structures. Slices viewed with maximum intensity projection. (**A**–**X**) are consequent images.

**Table 1 pharmaceutics-16-00478-t001:** Minimum inhibitory concentrations (MIC) and minimum bactericidal concentrations (MBC) of tested substances.

No.	Strain	RT	GO		PO		CU		CI		TCI	
			MIC % *v*/*v*	MBC % *v*/*v*	MIC % *v*/*v*	MBC% *v*/*v*	MIC µg/mL	MBC µg/mL	MIC % *v*/*v*	MBC % *v*/*v*	MIC % *v*/*v*	MBC % *v*/*v*
1	630	012	>50	>50	>50	>50	2048	4096	12.5	25	12.5	25
2	ATCC 9689	001	25	50	25	50	2048	4096	12.5	25	6.25	12.5
3	M120	078	>50	>50	>50	>50	2048	4096	12.5	25	12.5	25
4	4308/13	027	50	>50	6.25	12.5	256	512	6.25	12.5	6.25	12.5
5	25694/12	023	>50	>50	25	50	2048	4096	12.5	25	6.25	12.5
6	2628/12	176	12.5	25	12.5	25	128	256	12.5	25	12.5	25
7	CD 15	046	12.5	25	25	50	2048	4096	12.5	25	6.25	12.5
8	1128/06	017	6.25	12.5	25	50	64	128	12.5	25	12.5	25

Abbreviations: GO—ginger oil; PO—peppermint oil; CU—curcumin; CI—cinnamaldehyde; TCI—trans-cinnamaldehyde; RT—ribotype; *v*/*v*—volume/volume.

**Table 2 pharmaceutics-16-00478-t002:** Interpretation of biofilm images from confocal laser scanning microscopy.

Strain	Initial State	GO Treatment	PO Treatment	CU Treatment
Strain 630	Heterogeneous, thin, sparse biofilm with low 3D structure, no visible microaggregates (Figure 2A)	Homogeneous, dense, thin biofilm, very rare microaggregates, no loose cells (Figure 2B)	Same as GO Treatment (Figure 2C)	Thicker biofilm, numerous microaggregates, cells curled up (Figure 2D)
Strain ATCC 9689	Homogeneous, dense biofilm, high 3D structure, small amount of microaggregates (Figure 2G)	Highly aggregated, irregular biofilm with high 3D structure (Figure 2H)	Homogeneous biofilm, low 3D structure, very thin, no microaggregates (Figure 2I)	Sparse, heterogeneous biofilm, high 3D structure, distinct microaggregates, rounded cells (Figure 2J)
Strain M120	Thick, dense, regular biofilm, high 3D structure (Figure 2M)	Homogeneous, thin biofilm, low 3D structure, small number of aggregates, rounded cells (Figure 2N)	Thick biofilm, high 3D structure, numerous microaggregates, rounded cells (Figure 2O)	Very thin, irregular biofilm, low 3D structure, few microaggregates, rounded cells (Figure 2P)
Clinical strain 4308/13	Highly homogeneous, thin, regular biofilm, low 3D structure, no visible microaggregates (Figure 3S)	Thicker, irregular biofilm, distinct microaggregates, changed cell morphology (Figure 3T)	Homogeneous, thick, dense biofilm, altered cell morphology (Figure 3U)	Thin, sparse biofilm, low 3D structure, no visible microaggregates (Figure 3V)
Clinical strain 25694/12	Homogeneous, thin biofilm, low 3D structure, few microaggregates (Figure 3A)	Homogeneous, thin biofilm, small amount of microaggregates (Figure 3B)	Homogeneous, thin, dense biofilm, low 3D structure (Figure 3C)	Thicker, homogeneous biofilm, distinct microaggregates, fairly high 3D structure (Figure 3D)
Clinical strain 2628/12	Thin, regular biofilm, rare 3D architecture (Figure 3E)	Thicker, irregular biofilm, microaggregates, altered cells (Figure 3F)	Heterogeneous, thick biofilm, high 3D architecture, distinct microaggregates (Figure 3G)	Very thick, regular biofilm, high 3D architecture, numerous microaggregates (Figure 3H)
Clinical strain CD15	Uniform, thick, dense, regular biofilm with microaggregates (Figure 3M)	Thinner than control, regular, thin biofilm, small amount of microaggregates (Figure 3N)	Thin, irregular biofilm, low 3D architecture, no visible microaggregates (Figure 3O)	Homogeneous, thick, very dense biofilm, high 3D architecture, numerous microaggregates, very long cells (Figure 3P)
Clinical strain 1128/06	Regular, thin, dense biofilm, low 3D architecture (Figure 3S)	Thick, dense, regular biofilm, high 3D architecture, numerous microaggregates (Figure 3T)	Thin, regular biofilm, low 3D architecture (Figure 3U)	Homogeneous, very thick, dense biofilm, high 3D architecture, microaggregates, elongated cells (Figure 3V)

## Data Availability

The data that support the findings of this study are available from the corresponding author, D.W., upon reasonable request.

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
