# Peer review of "Antimicrobial Effects of Some Natural Products on Adhesion and Biofilm Inhibition of Clostridioides difficile"

_pharmaceutics, 2024, doi:10.3390/pharmaceutics16040478_

Round 1
Reviewer 1 Report
Comments and Suggestions for Authors
The paper of Dorota Wultanska, Michal Piotrowski and Hanna Pituch "Antimicrobial effects of natural compounds on adhesion and biofilm inhibition of Clostridioides difficile" is devoted to the in vitro study of bactericidal activity of five natural products against a Gram-positive bacterium C. difficile, which is causing diarrhea, colitis, hospital-related infections and others. Ginger oil (GO), peppermint oil (PO), curcumin (CU), cinnamon aldehyde (CI) and trans-cinnamaldehyde (TCI) were investigated on C. difficile adhesion and biofilm disruption. Tested substances showed bactericidal activity against various strains of C. difficile including hypervirulent strains. The manuscript is good written and may be of interest for researchers in the field of natural products, microbiology, pharmacy and pharmacology, antimicrobial drug design and development new agents for combating biofilm-related infections, but needs in some corrections.
1. The Title is very common for the presented data. it may be corrected as "Antimicrobial effects of some natural...".
2. The last sentence in Abstract should be improved. The studied natural products may be the subject for the extended studies as antimicrobial agents combating biofilm-related infections in the future according the presented data, and their use for the prevention and treatment of C. difficile infections is a long-term study. For today, there are a lot of natural products and individual compounds studying as the future antimicrobial agents.
3. Some corrections should be made in the text (see comments in PDF). In some places it is more suitable the words "natural products" instead of "natural compounds" because of authors studied oils too, and they are mixtures.
Reviewer

Author Response
Authors: Thank you for this review. Please see below for responses to your valuable comments
- The Title is very common for the presented data. it may be corrected as "Antimicrobial effects of some natural...".
Authors: Thank you for this suggestion, we have modified the title accordingly.
- The last sentence in Abstract should be improved. The studied natural products may be the subject for the extended studies as antimicrobial agents combating biofilm-related infections in the future according the presented data, and their use for the prevention and treatment of C. difficile infections is a long-term study. For today, there are a lot of natural products and individual compounds studying as the future antimicrobial agents.
Authors: Thank you for this suggestion, we have modified this sentence.
- Some corrections should be made in the text (see comments in PDF). In some places it is more suitable the words "natural products" instead of "natural compounds" because of authors studied oils too, and they are mixtures.
Authors: Thank you, we have changed this as suggested.
Reviewer 2 Report
Comments and Suggestions for Authors
The work shows an exhaustive investigation of different essential oils and compounds with recognized antimicrobial activity. Specifically, the effect on different strains of C. difficile was evaluated, in terms of their viability, cell adhesion and ability to form biofilms. The manuscript is interesting since it considers a greater breadth of strains. However, certain aspects must be clarified before considering its publication. The discussion section should delve into the possible mechanisms that explain what is reported in the results and not stop at the merely descriptive. The conclusion must also be rewritten.
The LSCM images provided in Figures 2 and 3 are complex to analyze, if possible, I suggest presenting them in another way. Furthermore, quantitative analyzes of biofilm spatial characteristics are missing. For example, with the images and the appropriate software, would it be possible to calculate the volume of biomass um3/um2; the average thickness (um), the roughness coefficient and the Surface to biovolume ratio? I think that would be a way to statistically compare the results and facilitate their understanding.
Specific comments:
• Figure 1. Indicate the degree of statistical significance of “*”
• Biofilm/biomass formation should be contrasted with a conventional method such as crystal violet/MTT, or SEM.
• It is desirable, although not mandatory, to include an antibiotic or DNAase as a control in experiments.
• Supplementary Figures S1-8 are presented visually unbalanced. There must be another way to show them
Comments on the Quality of English LanguageI did not detect major problems with the language in this work
Author Response
The work shows an exhaustive investigation of different essential oils and compounds with recognized antimicrobial activity. Specifically, the effect on different strains of C. difficile was evaluated, in terms of their viability, cell adhesion and ability to form biofilms. The manuscript is interesting since it considers a greater breadth of strains. However, certain aspects must be clarified before considering its publication. The discussion section should delve into the possible mechanisms that explain what is reported in the results and not stop at the merely descriptive. The conclusion must also be rewritten.
The LSCM images provided in Figures 2 and 3 are complex to analyze, if possible, I suggest presenting them in another way. Furthermore, quantitative analyzes of biofilm spatial characteristics are missing. For example, with the images and the appropriate software, would it be possible to calculate the volume of biomass um3/um2; the average thickness (um), the roughness coefficient and the Surface to biovolume ratio? I think that would be a way to statistically compare the results and facilitate their understanding.
Authors: Thank you for this review. We have more complex mechanisms of action of tested substances and changed conclusion section. We made our best effort to analyse biofilm images, but the images were saved and collected in TIF format. However, the available software is designed to analyze biofilm images that are saved in ND2 format in 3D.
Specific comments:
- Figure 1. Indicate the degree of statistical significance of “*”
Authors: Thank you, we have added this.
- Biofilm/biomass formation should be contrasted with a conventional method such as crystal violet/MTT, or SEM.
Authors: We attempted to observe biofilm formation by cultivating it on titration plates. However, this approach proved to be not productive as the substances interacted adversely with the material of the titration plates, worsen the effects. We have added this as a study limitation.
- It is desirable, although not mandatory, to include an antibiotic or DNAase as a control in experiments.
Authors: Unfortunately we do not have such an opportunity at the moment.
- Supplementary Figures S1-8 are presented visually unbalanced. There must be another way to show them
Authors: We agree, however, our program does not have the ability to show pairwise comparisons in any other way, so we put it in the supplement because they take up too much space.
Reviewer 3 Report
Comments and Suggestions for Authors
This manuscript reports the antimicrobial effects of natural compounds on the adhesion and biofilm inhibition of Clostridioides difficile. The natural compounds investigated were ginger oil, peppermint oil, curcumin, cinnamon aldehyde, and trans-cinnamaldehyde. The aims of the manuscript are interesting, considering the emergence of hypervirulent strains of C. difficile, and the work was apparently well conducted. However, there are some points that need explanation and/or correction. Please see the comments below.
1. The abstract should report better the results obtained, i. e., results should be organized to facilitate their understanding.
2. Keywords: Several keywords already appear in the title. Please replace these keywords by others such as Clostridium difficile, antibiofilm, antibacterial activity, plant bioactive compounds.
3. Introduction:
- Lines 36-43: please provide more detailed citations for the information described between lines 36 and 43;
4. Materials and Methods:
- Lines 76-77: please provide a better characterization of the three reference strains;
- Lines 77-79: please provide information on how the clinical isolates were confirmed as C. difficile;
- Lines 105-107: How were the concentrations tested chosen/defined? Were the authors based on any previous work or results? This information must be included in the methodology;
- Lines 156-157: “Confocal laser scanning microscopy according to the protocol described in previous studies [21, 22].” – please rephrase to: “Confocal laser scanning microscopy was performed according to the protocol described in previous studies [21, 22]”;
- Lines 161-169: the way the concentrations used are described is confusing. Please provide this information in a table. Also, why these concentrations were chosen/defined? This information must be included in the methodology;
- Lines 182-186: Statistical analysis - please provide the p-value considered significant;
5. Results
- Line 189: “...strains were susceptible to CU, with the MIC ranging from 128 µg/mL to 2048 µg/mL...” – please check this information – wouldn't it be 64 to 2048 µg/mL (according Table 1)?
- Table 1: please rephrase the caption of Table 1 to: “Table 1. Minimum Inhibitory Concentrations (MIC) and Minimum Bactericidal Concentrations (MBC) of tested substances”;
- Table 1: “Abbreviations: GO – ginger oil; PO – peppermint oil; CU – curcumin; CI – cinnamaldehyde; TCI - trans-cinnamaldehyde; MIC – minimum inhibitory concentration; MBC – minimum bactericidal concentration; RT – ribotype; v/v – volume/volume - please rephrase to: “GO – ginger oil; PO – peppermint oil; CU – curcumin; CI – cinnamaldehyde; TCI - trans-cinnamaldehyde; RT – ribotype; v/v – volume/volume “;
- Lines 209-210: “GO, CU, and TCI significantly inhibited the adhesion of strain 630 to CCD841 com cells” – please check this information since Figure 1 does not show this;
- Lines 210 – 211: “All substances showed a very high capacity to inhibit the adhesion of C. difficile strain 630 to HT-29 MTX cells” - please check this information since Figure 1 does not show this;
- Lines 206-223: please rephrase the way these results are described;
- Figure 1:
a) please review this figure - display all graphs with the y-axis on the same scale;
b) use very different colors in the bars that represent cell lines;
- Lines 229-288: please review the way you describe these results - the description is not fluent and is confusing. Organize these results better;
- Figures 2 e 3:
a) please replace the numbers shown within the figures to letters;
b) please improve the quality of the scale bars shown in the Figures 2 and 3 and add in the legends what magnification they represent;
c) please review the captions of Figures 2 and 3 – the caption must be self-explanatory – provide the meaning of the abbreviated terms shown in the figures and provide as much information as possible about the data;
6. Conclusion: the conclusion presents a summary of the results. It should finalize the findings presented and point out perspectives for the advancement of knowledge in the area studied – please review.
In my final comments, I recommend that the manuscript should be widely reviewed by the authors. The abstract, material and methods, results and conclusion sections must be rephrased to value the results obtained and the scientific advances for the area. The manuscript should be widely revised by a native English speaker.
Comments on the Quality of English LanguageExtensive editing of English language is required.
Author Response
- The abstract should report better the results obtained, i. e., results should be organized to facilitate their understanding.
Authors: Thank you for this suggestion, we have changed the abstract.
- Keywords: Several keywords already appear in the title. Please replace these keywords by others such as Clostridium difficile, antibiofilm, antibacterial activity, plant bioactive compounds.
Authors: Thank you for this suggestion, we have proposed new keywords.
- Introduction:
- Lines 36-43: please provide more detailed citations for the information described between lines 36 and 43;
Authors: Thank you, we have changed it.
- Materials and Methods:
- Lines 76-77: please provide a better characterization of the three reference strains;
Authors: We have provided toxigenicity of these strains.
- Lines 77-79: please provide information on how the clinical isolates were confirmed as C. difficile;
Authors: We have described how the clinical isolates were confirmed.
- Lines 105-107: How were the concentrations tested chosen/defined? Were the authors based on any previous work or results? This information must be included in the methodology;
Authors: In order to determine the MIC and MBC values, the concentrations of the tested substances were serially diluted from 50% v/v stock solution.
- Lines 156-157: “Confocal laser scanning microscopy according to the protocol described in previous studies [21, 22].” – please rephrase to: “Confocal laser scanning microscopy was performed according to the protocol described in previous studies [21, 22]”;
Authors: We have changed this.
- Lines 161-169: the way the concentrations used are described is confusing. Please provide this information in a table. Also, why these concentrations were chosen/defined? This information must be included in the methodology;
Authors: Thank you for this suggestion; we have removed these concentrations and clarified that they were just half of the MIC.
- Lines 182-186: Statistical analysis - please provide the p-value considered significant;
Authors: We have added such information.
- Results
- Line 189: “...strains were susceptible to CU, with the MIC ranging from 128 µg/mL to 2048 µg/mL...” – please check this information – wouldn't it be 64 to 2048 µg/mL (according Table 1)?
Authors: True, we have changed this.
- Table 1: please rephrase the caption of Table 1 to: “Table 1. Minimum Inhibitory Concentrations (MIC) and Minimum Bactericidal Concentrations (MBC) of tested substances”;
Authors: Thank you, we have modified table caption.
- Table 1: “Abbreviations: GO – ginger oil; PO – peppermint oil; CU – curcumin; CI – cinnamaldehyde; TCI - trans-cinnamaldehyde; MIC – minimum inhibitory concentration; MBC – minimum bactericidal concentration; RT – ribotype; v/v – volume/volume - please rephrase to: “GO – ginger oil; PO – peppermint oil; CU – curcumin; CI – cinnamaldehyde; TCI - trans-cinnamaldehyde; RT – ribotype; v/v – volume/volume “;
Authors: Thank you, we have changed it.
- Lines 209-210: “GO, CU, and TCI significantly inhibited the adhesion of strain 630 to CCD841 com cells” – please check this information since Figure 1 does not show this;
Authors: True, it should be increased.
- Lines 210 – 211: “All substances showed a very high capacity to inhibit the adhesion of C. difficile strain 630 to HT-29 MTX cells” - please check this information since Figure 1 does not show this;
Authors: Please check it again, HT-29 MTX cell line is marked in green now.
- Lines 206-223: please rephrase the way these results are described;
Authors: We have rephrased these results.
- Figure 1:
- a) please review this figure - display all graphs with the y-axis on the same scale;
- b) use very different colors in the bars that represent cell lines;
Authors: We have introduced suggested changes.
- Lines 229-288: please review the way you describe these results - the description is not fluent and is confusing. Organize these results better;
Authors: We have introduced a table to provide a better description of the results.
- Figures 2 e 3:
- a) please replace the numbers shown within the figures to letters;
- b) please improve the quality of the scale bars shown in the Figures 2 and 3 and add in the legends what magnification they represent;
- c) please review the captions of Figures 2 and 3 – the caption must be self-explanatory – provide the meaning of the abbreviated terms shown in the figures and provide as much information as possible about the data;
Authors: We have modified figures accordingly. We were not able to change quality of scale bars, we have added description in legend instead. Therefore, we have added high-quality images to the supplement because there are too large to be included in main file.
- Conclusion: the conclusion presents a summary of the results. It should finalize the findings presented and point out perspectives for the advancement of knowledge in the area studied – please review.
Authors: We have changed entire paragraph.
In my final comments, I recommend that the manuscript should be widely reviewed by the authors. The abstract, material and methods, results and conclusion sections must be rephrased to value the results obtained and the scientific advances for the area. The manuscript should be widely revised by a native English speaker.
Authors: Thank you for this opinion, we sent the work for proofreading by a native speaker and he made corrections, we hope that now the language meets expectations.
Round 2
Reviewer 2 Report
Comments and Suggestions for Authors
This new version of the manuscript was significantly improved. In my opinion, the main doubts and observations were answered adequately and clearly by the author.
Reviewer 3 Report
Comments and Suggestions for Authors
Please check the references. References are not presented in accordance with the format recommended in the instructions for authors. Please review.